# Generalizing ISP Model by Unsupervised Raw-to-raw Mapping

Anonymous Authors

## ABSTRACT

ISP (Image Signal Processor) serves as a pipeline converting unprocessed raw images to sRGB images, positioned before nearly all visual tasks. Due to the varying spectral sensitivities of cameras, raw images captured by different cameras exist in different color spaces, making it challenging to deploy ISP across cameras with consistent performance. To address this challenge, it is intuitively to incorporate a raw-to-raw mapping (mapping raw images across camera color spaces) module into the ISP. However, the lack of paired data (i.e., images of the same scene captured by different cameras) makes it difficult to train a raw-to-raw model using supervised learning methods. In this paper, we aim to achieve ISP generalization by proposing the first unsupervised raw-to-raw model. To be specific, we propose a CSTPP (Color Space Transformation Parameters Predictor) module to predict the space transformation parameters in a patch-wise manner, which can accurately perform color space transformation and flexibly manage complex lighting conditions. Additionally, we design a CycleGAN-style training framework to realize unsupervised learning, overcoming the deficiency of paired data. Our proposed unsupervised model achieved performance comparable to that of the state-of-the-art semi-supervised method in raw-to-raw task. Furthermore, to assess its ability to generalize the ISP model across different cameras, we for the first formulated cross-camera ISP task and demonstrated the performance of our method through extensive experiments. Codes will be publicly available.

## CCS CONCEPTS

• **Computing methodologies → Computational photography**; **Image processing**; *Computer vision problems*;

## KEYWORDS

Image Signal Processor, Unsupervised Raw-to-raw Mapping, Parameterized Model

## 1 INTRODUCTION

ISP refers to an image processing system aiming to transform the unprocessed raw images captured by camera into visually pleasing sRGB images for human perception, while serving for various downstream visual tasks [24, 26, 27, 32, 33, 35, 37, 39], such as object detection, semantic segmentation, and instance segmentation. The system of a traditional ISP includes white balance module, denoising module, demosaicing module etc. Since the features are manually designed, traditional ISP suffers from problems such as

**Unpublished working draft. Not for distribution.**

Permission to make digital or hard copies of all or part of this work for personal or classroom use is granted without fee provided that copies are not made or distributed for profit or commercial advantage and that copies bear this notice and the full citation on the first page. Copyrights for components of this work owned by others than the author(s) must be honored. Abstracting with credit is permitted. To copy otherwise, or republish, to post on servers or to redistribute to lists, requires prior specific permission and/or a fee. Request permissions from permissions@acm.org.

*ACM MM, 2024, Melbourne, Australia*

© 2024 Copyright held by the owner/author(s). Publication rights licensed to ACM.
ACM ISBN 978-x-xxxx-xxxx-x/YY/MM
https://doi.org/10.1145/nnnnnnn.nnnnnnn

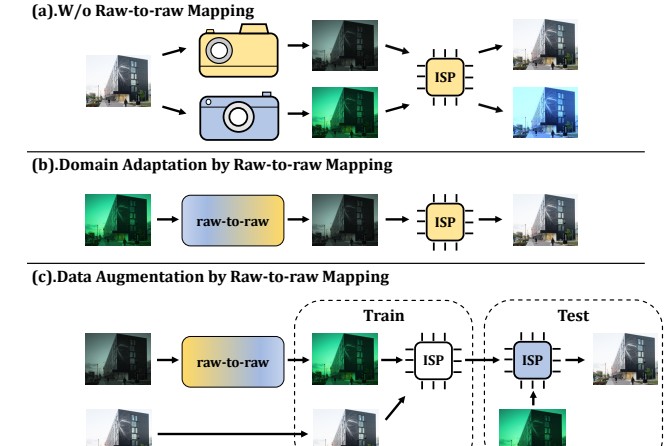

**Figure 1: Generalizing ISP model by raw-to-raw mapping. Top shows the challenge of ISP's cross-camera deployment, middle shows the application of raw-to-raw in the test phase (i.e., domain adaptation), bottom shows the application of raw-to-raw in the train phase (i.e., data augmentation).**

tedious parameter tuning, low flexibility and limited adaptability [12, 18]. In recent years, with the rapid advancement of deep learning technology, AI-ISP has emerged as a more effective approach for raw image processing. Current AI-ISP methods either replace sub-modules in the traditional ISP system [1, 3, 5, 6, 11] or the entire ISP[15–18] with an end-to-end deep neural network. By training on raw images that are paired with expert fine-tuned or professional camera rendered sRGB images, AI-ISP achieves much higher image quality than a traditional ISP.

However, owing to the varying spectral sensitivities of cameras, raw images captured by different cameras exist in different color spaces, caused AI-ISP exhibit poor performance when deployed across different cameras. **Fig.** 1: (a) presents an example to illustrate it, when deployed on a blue camera, the yellow ISP fails to render the sRGB image accurately. While some cross-camera methods and multi-camera datasets have been developed for tasks such as white balance [1, 3, 4, 7] and denoising [8, 14, 38], research on the cross-camera deployment of entire ISP models remains limited.

Intuitively, a raw-to-raw mapping module, designed to convert raw images across camera color spaces, can be placed before the ISP to ensure consistency in the raw data from different cameras, as illustrated in **Fig.** 1: (b) (c). Previous raw-to-raw methods [2, 28] require physical access to cameras and generate pixel-level-paired images from different cameras for training, which is a tedious and difficult process. In addition to this, their methods assume uniform illumination condition, which may not suitable in scenes with complex lighting.

Unpaired image translation provides an alternative solution to the raw-to-raw task. However, the inherent domain gaps present in unpaired images taken by different cameras, stemming from

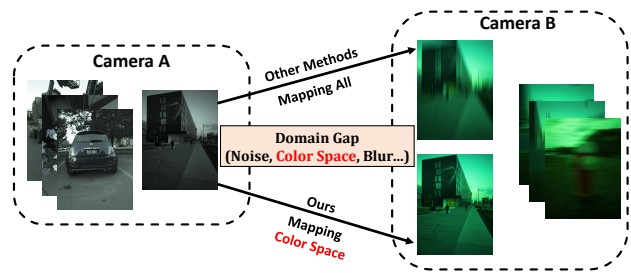

**Figure 2: Multiple domain gap issues in the raw-to-raw task.**

variations in noise, blur, and camera physical characteristics, pose a significant challenge for raw-to-raw tasks under unsupervised settings as illustrated in **Fig.** 2. Traditional unsupervised image translation methods may struggle when faced with uncertain domain gaps, potentially resulting in erroneous mappings.

To address the above-mentioned challenges, we propose a novel unsupervised raw-to-raw model which is trained based on the CycleGAN framework [41]. A key component of our model named CSTPP (Color Space Transformation Parameters Predictor) predicts the color space transformation parameters and applies them in a patch-wise manner, rather than performing per-pixel image translation. This parameterized method allows our model to focus on the domain gap of the color space while ignoring others it cannot map, thereby enabling correct color space transformation. Our proposed model serves dual purposes: it can function as a domain adaptation module during the testing phase of the ISP model, and it can also be used as a data augmentation method during the training phase as illustrated in **Fig.** 1: (b) (c), respectively. In summary, the contributions of this paper are as follows:

- We propose an unsupervised raw-to-raw model for the first time. In the raw-to-raw task, our unsupervised method performs equally well as Afifi et al.'s semi-supervised method [2], surpassing significantly the results of other traditional unsupervised methods.
- We have extended the raw-to-raw task to multi-illumination scenes and achieved the best results in corresponding quantitative evaluation.
- In the proposed cross-camera ISP task, we conducted extensive quantitative and qualitative evaluations. Despite the absence of paired training data from target camera, our method still enables the ISP model to produce visually pleasing results.

## 2 RELATED WORK

### 2.1 ISP

Traditional ISP typically consists of modules for denoising, demosaicing, white balancing, color toning etc. Experts need to spend considerable time carefully tuning parameters for each module. With the development of deep learning, end-to-end networks have been introduced to directly reconstruct sRGB images from raw images. Ignatov et al. [18] first proposed a high-quality raw-rgb dataset, consisting of pairs of raw images captured by a mobile phone camera and sRGB images rendered by a professional camera. They accomplished this challenging reconstruction task using a

pyramid model that trained hierarchically. With the demand for deploying AI-ISP into mobile devices, current state-of-the-art AI-ISP models, such as syenet[15] and microISP[17], not only achieve high-quality results but also have few parameters.

In addition to serving human vision, some researchers focus on the impact of ISP on downstream vision tasks [24, 26, 27, 32, 33, 35, 37, 39] such as object detection, semantic segmentation, and instance segmentation. Mosleh et al. [27] differentiated the ISP modules and deployed them in a series of downstream tasks. They optimized the parameters of the ISP module end-to-end by incorporating the loss from downstream tasks. Sun et al. [35] introduced reinforcement learning into ISP task, they dynamically adjust the parameters of the ISP module based on the performance of downstream tasks. Compared to traditional ISP designed for human vision, their ISP for machine perception can achieve better performance on a variety of visual tasks.

While ISP has found success in various imaging systems, the majority of current ISP algorithms are customized for specific cameras. Significant domain gaps between raw images captured by different cameras pose challenges for deploying ISP across various cameras [10]. In general, each camera requires a separate dataset for ISP model training (learning-based ISP) or extensive parameter tuning (traditional ISP). Additionally, training ISP using joint multi-camera raw data is challenging due to the lack of paired data and the difficulty for models to simultaneously map from multiple camera-specific color spaces to sRGB.

### 2.2 Raw-to-raw

The objective of the raw-to-raw task is to establish a mapping function $f$ capable of accurately transforming raw images from Camera A's color space to Camera B's color space, accommodating diverse scenes and lighting conditions. In brief, the mapping function can be expressed as:

$$I^B = f(I^A), \qquad (1)$$

where $I^A$ and $I^B$ denote the raw images captured by camera A and camera B, respectively, under identical lighting and scene conditions.

From [20, 28], the raw-to-raw mapping can be approximated as a channel-wise color transformation $T(L) \in \mathbb{R}^{3\times3}$. Under illumination condition $L$, the packed raw image $I^A = [r, g, b]$ captured by camera A can be transformed to the color space of camera B approximately as follows:

$$I^B = f(I^A, L) \approx I^A T(L). \qquad (2)$$

Nguyen et al. [28] proposed that by employing irreversible quadratic transformation, equation 2 can be further approximated by **Eq.** 3:

$$f(I^A, L) \approx I^A_{qt} T_{qt}(L), \qquad (3)$$

here, $I^A_{qt} = [r^2, g^2, b^2, r \times g, g \times b, r \times b, r, g, b]$, $T_{qt} \in \mathbb{R}^{9\times3}$. Correspondingly, for raw images with four channels $[r, g_r, g_b, g]$ rather than three, it can simply generalize that $I^A_{qt} = [r^2, g_r^2, \cdots, g_b, b]$ and $T_{qt} \in \mathbb{R}^{14\times4}$.

Koskinen et al. [22] and C5 [3] attempted to enhance the cross-camera generalization of the white balance module using raw-to-raw mapping. However, their methods require ground truth illumination and extensive meta data. Afifi et al. [2] were the first to

introduce neural networks into the raw-to-raw task, they placed a standard color chart in the scene and captured raw images using two cameras. Subsequently, they estimated the parameters of the quadratic transformation $T_{qt}$ by minimizing the color differences between the two color charts. They applied this transformation to the raw images and obtaining pixel-level-paried data. They then utilized a Unet-like [34] model for semi-supervised training. Although their trained model can accomplish raw-to-raw mapping for other scenes without having to again capture paired images, access to actual devices to create paired dataset before training is still inevitable. Moreover, this approach is only suitable for ideal scenes with uniform lighting. In general scenes with non-uniform lighting, the transformation parameters vary significantly across different regions of the image, making it impossible to produce high-quality supervised data and conduct supervised training.

## 2.3 Unpaired Image Translation

The goal of the UNIT (Unpaired Image Translation) task is to map images from domain $\chi^A$ to domain $\chi^B$ without ground truth, such as from sunny to rainy or from male to female. Zhu et al. [41] proposed CycleGAN for the UNIT task, utilizing cycle-consistency loss and identity loss to bridge domain gaps. Cut [29] introduced a contrastive learning approach to image translation task and achieved good results in cases where only one-way mappings exist. UVC-GAN [36] introduced a pixel-wise transformer into the CycleGAN framework, while Swin-UNIT [23] addressed performance issues caused by high-resolution images by introducing swin-transformer block which has a linear complexity.

Unlike the per-pixel translation methods mentioned above, Chai et al [9]. proposed a parameterized network for predicting color enhancement parameters. Their method achieved excellent visual results on the MIT-5K color enhancement dataset. However, their method applies a uniform transformation across the entire image, resulting in limited flexibility in tasks calling for local transformation.

## 3 METHOD

### 3.1 Patch-wise Color Space Transformation

In this section, we discuss patch-wise color space transformation. It is worth noting that the color space transformation mentioned here refers to the direct conversion from the color space of one camera to the color space of another camera. The pixel values of raw images can be directly modeled as follows [28]:

$$I^{cam} = \int_a^b R(\lambda)L(\lambda)C^{cam}(\lambda)d\lambda, cam \in \{A, B\}, \quad (4)$$

here, $[a, b]$ represents the visible light wavelength range 380nm-720nm, $R(\lambda)$ represents the spectral reflectance property at wavelength $\lambda$, $L(\lambda)$ denotes the spectral intensity at wavelength $\lambda$, and $C^{cam}(\lambda)$ denotes the camera sensor sensitivity at wavelength $\lambda$.

By discretizing the integral form of **Eq.** 4, an equivalent matrix representation can be obtained, as shown in **Eq.** 5:

$$I_{1\times 3}^{cam} = R_{1\times n}L_{n\times n}C_{n\times 3}^{cam}, cam \in \{A, B\}, \quad (5)$$

here, $n$ represents the discretized resolution. $R$ is a vector composed of the reflectance at various wavelengths. $L$ is a diagonal matrix

where the elements on its diagonal represent the spectral power of the scene illumination at each wavelengths. Each columns of $C^{cam}$ represents the sensitivity curve of the red, green, and blue filters respectively (we approximate filters within a Bayer unit to be located at the same position). For a raw image with m pixels, when applying a single transformation, the raw-to-raw task can be seen as solving the following matrix equation for the unknown matrix $T_{3\times 3}$:

$$\begin{cases} R^1 L^1 C^A T = R^1 L^1 C^B \\ R^2 L^2 C^A T = R^2 L^2 C^B \\ \quad \vdots \\ R^m L^m C^A T = R^m L^m C^B. \end{cases} \quad (6)$$

Since both $C^A$ and $C^B$ are not invertible square matrices, there is no exact solution $T$ that holds for any $R$ and $L$ [20]. The method proposed by Nguyen et al. [28] essentially involves finding an approximate solution $T(L)$ for each lighting condition $L$, assuming $R^1, R^2, \cdots, R^m$ are standard color chart reflectances. While this approach has achieved extremely low color chart errors, it still faces the following problems: Firstly, attempting to approximate multiple color mappings with single $T(L)$ may lead to significant discrepancies in the elements of $T(L)$, making it more likely to become an ill-conditioned matrix. This, in turn, can result in certain parts of the image being mapped outside the color gamut [2]. Secondly, applying the same transformation $T(L)$ (i.e., Global calibration mentioned in **Table** 1) to multiple raw images captured under different lighting conditions yields poor results. From this, we can infer that globally applying the same transformation $T(L)$ in a non-uniform lighting scene could degrade performance. Additionally, for neural networks, learning the mapping from images to ill-conditioned matrices is much more challenging compared to learning the mapping from images to regular matrices.

To address these issues, we propose predicting transformation parameters in a patch-wise manner. Within a small patch, reflectance and lighting condition are roughly consistent, implying that **Eq.** 6 only involves a single row. Patch-wise color space transformation can be seen as solving multiple independent single-row equations as illustrated in **Eq.** 7:

$$\begin{cases} R^1 L^1 C^A T^1 = R^1 L^1 C^B \\ R^2 L^2 C^A T^2 = R^2 L^2 C^B \\ \quad \vdots \\ R^m L^m C^A T^m = R^m L^m C^B. \end{cases} \quad (7)$$

Solving these equations for an approximate solution is much easier for neural networks, and it is also less likely to result in mapping outside the color gamut.

### 3.2 Overall Framework

Our proposed model is training based on the CycleGAN architecture [41]. CycleGAN is a general unsupervised image-to-image translation framework that learns mutual mappings between domain $\chi^A$ and domain $\chi^B$ by jointly training two pairs of generators and discriminators. The overall framework is illustrated in **Fig.** 3. In the

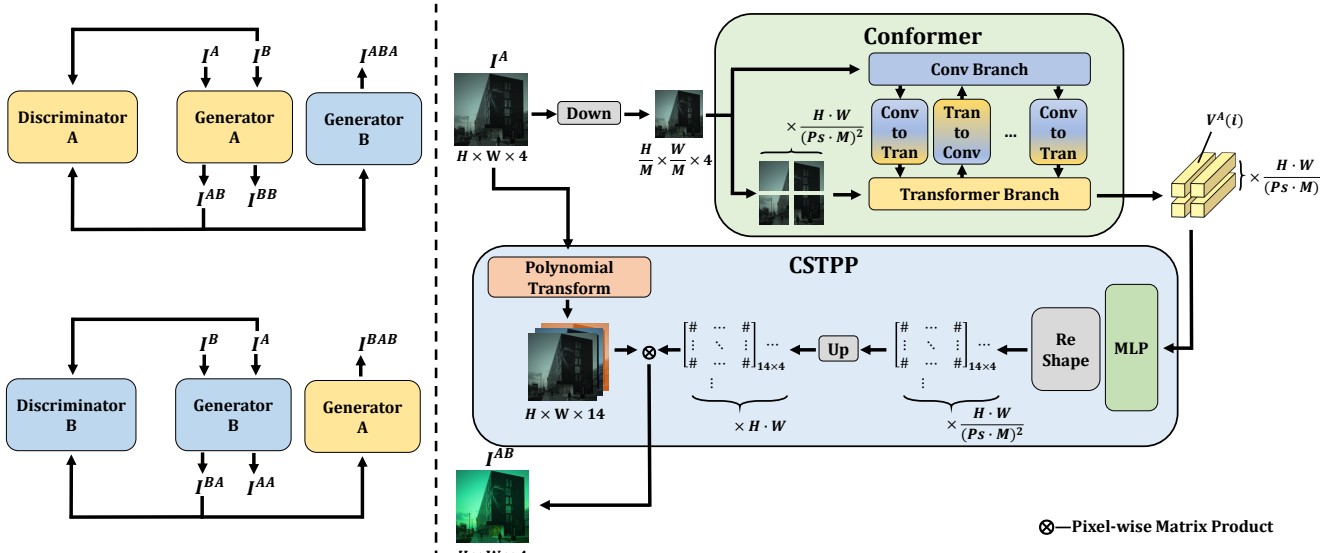

**Figure 3: Our proposed method. Left is the architecture of CycleGAN. Right is the specific structure of proposed generator. The shape of each element in the figure corresponds to the situation that the input raw image has four channels (packed rggb) and the quadratic transformation is applied.**

raw-to-raw task, $\chi^A$ represents the color space of camera A, while $\chi^B$ represents the color space of camera B. Generators are trained to map raw images between color spaces A and B while preserving other information such as texture, scene and lighting information. Discriminators play a role in discerning between generated images and real ones, guiding generators to produce more authentic outputs. For the discriminator, we employ the same structure as the discriminator in [41]. For the generator, we propose a novel structure based on the CSTPP module, which will be discussed in **Sec.** 3.3.

During training, the total loss function for the two generators $Loss_{gen}$ consists of three components. The cycle consistency loss, as shown in **Eq.** 8, is employed to maintain the structural integrity of the raw image during the raw-to-raw mapping:

$$Loss_{cycle} = \|I^A - I^{ABA}\|_1 + \|I^B - I^{BAB}\|_1, \qquad (8)$$

here, $I^A$ represents the raw image captured by camera A, $I^{AB}$ denotes the output when $I^A$ is attempted to be transformed to the color space of camera B by the generator, and $I^{ABA}$ represents the result when this output is fed into another generator, attempting to transform it back to the color space of camera A.

Since raw images are always in a specific camera color space, we apply the identity loss as shown in **Eq.** 9, to encourage the generator to discern the color space of the raw images and to perform an identity mapping on raw images that belong to the same color space:

$$Loss_{iden} = \|I^A - I^{AA}\|_1 + \|I^B - I^{BB}\|_1, \qquad (9)$$

The adversarial loss, as shown in **Eq.** 10, is employed to optimize the generators under the guidance of discriminators. It's worth noting that we utilize the lsgan loss [25] instead of the cross-entropy adversarial loss, which facilitates more stable training:

$$Loss_{gan} = (1 - Dis^A(I^{AB}))^2 + (1 - Dis^B(I^{BA}))^2, \qquad (10)$$

here, $Dis(*)$ represents the probability output by the discriminator. The total loss function for the generator is depicted in **Eq.** 11:

$$Loss_{gen} = \alpha Loss_{cycle} + \beta Loss_{iden} + Loss_{gan}, \qquad (11)$$

here, $\alpha$ and $\beta$ represent hyperparameters, which are set to 10 and 5 in the following experiments, respectively. The total loss function for the discriminator is given by **Eq.** 12:

$$Loss_{dis} = (Dis^A(I^{AB}))^2 + (Dis^B(I^{BA}))^2$$
$$+ (1 - Dis^A(I^B))^2 + (1 - Dis^B(I^A))^2. \qquad (12)$$

### 3.3 Raw-to-raw Mapping Network

In recent years, models based on ViT [13] have seen widespread adoption in the field of computer vision. Benefiting from their powerful capability to extract global information, ViT-based models have achieved remarkable success in many visual tasks. As a variant of ViT, Conformer [31] includes both transformer and convolution branch. While the transformer branch extracts global features, the convolution branch effectively captures local features. Moreover, the two branches exchange information at various levels through upsampling and downsampling mechanisms.

As described in **Sec.** 3.1, our proposed model needs to predict the color space transformation parameters for each patch based on both reflectance and lighting information. Since RGB values are determined by both lighting and object reflectance, the same RGB value may correspond to multiple combinations of lighting and reflectance. Therefore, relying solely on information within the patch cannot separate the two, transformation parameters heavily reliant on global information. Based on this, we use a conformer as the backbone of our generator to extract local and global features corresponding to each patch, which are then mapped to polynomial transformation parameters by the CSTPP module.

In the situation that the input raw image has four channels (packed rggb) and applying quadratic transformation, raw image

$I^A$ captured by camera A undergoes downsampling by a factor of $M$. Subsequently, $I^A$ will be input into the convolution branch, at the same time, $I^A$ will be splited into several patches $I^A(i)$ of size $Ps{\times}Ps$, and soon be fed into the transformer branch. Conformer acting as a feature extractor, processes each $I^A(i)$ into feature vectors $V^A(i) \in \mathbb{R}^{dim}$ containing both local and global information, where $Dim$ is the embedding dimension of transformer branch. After that, the CSTPP module processes the image as follows:

$$I^A_{qt} = PT(I^A),$$

$$T_{qt}(i) = ReShape(MLP(V^A(i))), i \in \{0, 1, \cdots, \frac{H \cdot W}{(Ps \cdot M)^2}\}, \quad (13)$$

$$T_{qt} = Up(T_{qt}(i)),$$

$$I^{AB} = I^A_{qt} T_{qt},$$

here, $PT$ refers to Polynomial Transform, and $Up$ denotes upsampling with bilinear interpolation applied to each parameter matrix $T_{qt}(i)$. The interpolation occurs between adjacent patch to prevent block boundary effects. Subsequently, $I^A$ will undergo pixel-wise matrix product with the upsampled parameter matrix, and the resulting raw image $I^{AB}$ in camera B's color space will be the output.

The CSTPP module limits the operations that the model can perform on raw image to channel-level polynomial combination, which typically only adjust the color of the image. This transformation ignores domain gaps caused by factors such as scenes, noise, blur, and physical properties of the camera, while effectively addressing what caused by the camera's spectral sensitivity(i.e. color space) as illustated in **Fig.** 2. Additionally, since we apply patch-wise transformation rather than global transformation, our model retains considerable flexibility. Even for scenes with complex lighting, our model can effectively perform raw-to-raw task.

## 4 EXPERIMENT

In subsequent experiments, we use the same model architecture. For the transformer branch, we set the embedding dimension to 256, patch size to 16, head to 4, and depth to 6. For the convolution branch, we set the base channels to 32 and gradually increase to 256 with the increase of depth. At the same time, we apply the same parameter sharing mechanism as Chai et al. [9] to stabilize the training process, this means that our generator A and generator B will share all parameters in the conformer module except for batch normalization layer.

We will conduct experiments on the raw-to-raw task under both single illumination and multi illumination conditions, respectively, to demonstrate the effectiveness of our parameterized method and the patch-wise strategy. We will also conduct experiments on the proposed cross camera ISP task to demonstrate the benefits of our method for ISP generalization.

### 4.1 Single Illumination Raw-to-raw Mapping

**Dataset.** For the evaluation of single illumination raw-to-raw task, we use the dataset proposed by Afifi et al. [2]. This dataset consists of 392 unpaired images captured by Samsung-s9 and iPhone-x, along with 137 pairs of pixel-level paired images created using the method described in **Sec.** 2.2. The paired images are further divided into an anchor set of 22 pairs used for training Afifi et al.'s model

and a test set of 115 pairs. Each raw image has a corresponding rendered sRGB image.

**Experiment Settings.** For this part of the experiment, the batch size of our model is set to 4. We use Adam optimizer [21] with betas set to 0.5 and 0.999. The learning rate for the generators are set to 1e-5, and for the discriminators are set to 2e-5. Other methods are trained using the same settings as described in their respective papers or codes. The dataset is split into $256 \times 256$ patches as mentioned in [2]. Random flipping and rotation are applied for data augmentation during training for 50 epochs. Except for the Afifi et al.'s model, which uses an additional 22 pixel-level paired images for training, all other models only use the 392 unpaired images for training. All models are tested on 115 pairs of full solution pixel-level paired images.

**Testing Results.** The quantitative results in terms of PSNR, SSIM, and MAE (Mean Absolute Error) metrics are shown in **Table** 1. Our unsupervised model achieves performance that comparable to Afifi et al.'s semi-supervised model across all metrics. By applying Chai et al.'s method, which is a parameterized model for color enhancement tasks, to the raw-to-raw task, it outperforms several other non-parameterized unsupervised models in SSIM metric, indicating that parameterized models can work better in preserving structural information in images. Raw-to-raw tasks precede ISP or other raw-inputting tasks, so introducing noise, deformation, or artifacts that may harm these tasks is detrimental. Our parameterized model effectively ensures that these issues will not occur, this can be reaffirmed by the qualitative results depicted in **Fig.** 4, where we achieved results closest to the ground truth, while non-parameterized unsupervised methods like Swin-UNIT (second column) and UVCGAN (fourth column) show varying degrees of degradation in the information present in the input raw images.

### 4.2 Multi Illumination Raw-to-raw Mapping

**Dataset.** We utilize the LSMI (Large Scale Multi Illumination) dataset [19] to assess the model's raw-to-raw mapping capability under non-uniform lighting conditions. The LSMI dataset comprises over 7486 raw images captured in more than 2700 scenes with multiple illuminations by three different cameras: Sony, Galaxy, and Nikon. Each image contains three standard color charts placed at different positions. Because there are overlapping scenes captured by both Sony and Galaxy, we utilized a total of 232 pairs of such scene-level paired images for testing, the remaining 3416 unpaired images are employed for training.

**Experiment Settings.** In this part of the experiment, the images are preprocessed following [19], other training settings are identical to those in **Sec.** 4.1.

Since the test set matches at the scene level rather than the pixel level, we use the KL Divergence Distance of the image pixel value distribution as the metric following [28]. This metric is used to evaluate the consistency of the overall color distribution of the image as illustrated in **Eq.** 14:

$$KL(I, \hat{I}) = \sum_{i \in bins} Count(I, i)ln(\frac{Count(I, i) + eps}{Count(\hat{I}, i) + eps})$$
$$+ \sum_{i \in bins} Count(\hat{I}, i)ln(\frac{Count(\hat{I}, i) + eps}{Count(I, i) + eps}), \quad (14)$$

**Table 1: Quantitative results of single illumination raw-to-raw mapping task. Bold indicates the best result. * indicates data sourced from [2].**

| Training Method | Model | Samsung-s9→iPhone-x | | | iPhone-x→Samsung-s9 | | |
|---|---|---|---|---|---|---|---|
| | | PSNR↑ | SSIM↑ | MAE↓ | PSNR↑ | SSIM↑ | MAE↓ |
| Non-learning | *Global calibration(3 × 3) [28] | 24.52 | 0.71 | 0.049 | 17.03 | 0.51 | 0.16 |
| | *Global calibration(poly) [28] | 24.88 | 0.72 | 0.048 | 16.88 | 0.50 | 0.16 |
| | *FDA [40] | 20.95 | 0.48 | 0.06 | 19.18 | 0.47 | 0.090 |
| Unsupervised | Cyclegan [41] | 24.55 | 0.76 | 0.046 | 25.21 | 0.76 | 0.042 |
| | Cut [29] | 23.51 | 0.71 | 0.050 | 22.44 | 0.71 | 0.053 |
| | Swin-UNIT [23] | 23.92 | 0.72 | 0.057 | 23.77 | 0.75 | 0.051 |
| | Chai et al.'s [9] | 29.35 | 0.86 | 0.028 | 27.78 | 0.86 | 0.037 |
| | UVCGAN [36] | 27.22 | 0.82 | 0.031 | 26.10 | 0.79 | 0.037 |
| | Ours | **29.73** | **0.90** | **0.025** | 28.09 | 0.89 | **0.033** |
| Semi-supervised | Afifi et al.'s [2] | 29.65 | 0.89 | 0.027 | **28.58** | **0.90** | **0.033** |

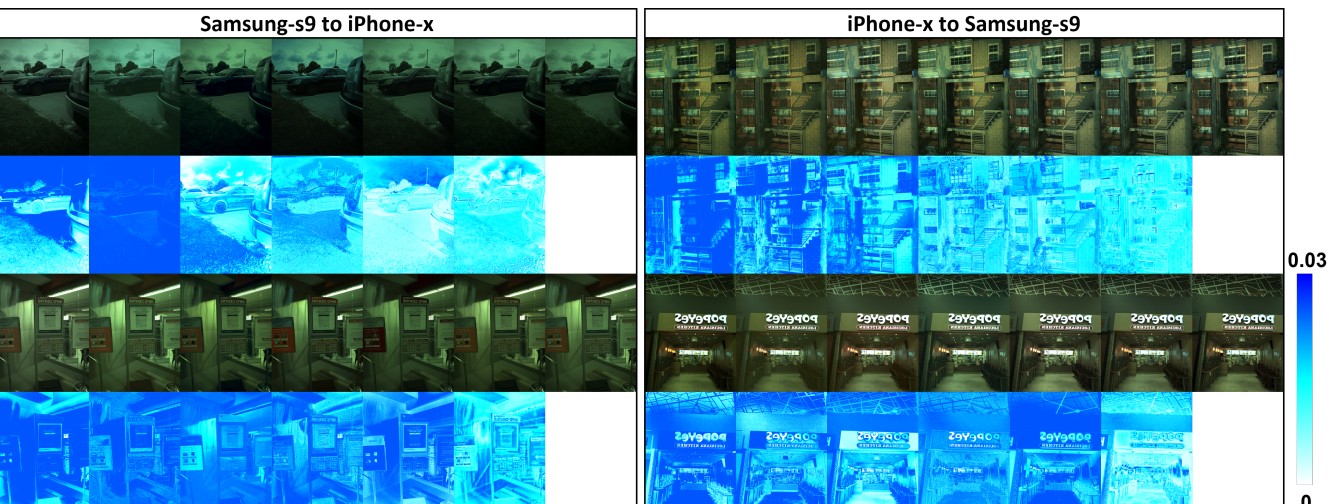

**Figure 4: Qualitative results of the single illumination raw-to-raw task. Each block, from left to right, represents the source camera, Swin-UNIT [23], Chai et al.'s [9], UVCGAN [36], Afifi et al.'s [2], our method, and ground truth, respectively. Odd rows show raw images with a 1/1.6 gamma correction, while even rows display the absolute errors between predicted images and ground truth.**

here, $bins = \{0, 1, \cdots, 255\}$ represents the set of pixel values, and $eps$ denotes a very small value to prevent division by zero or $ln(0)$, which we set as 1e-7. It should be noted that we calculate the KL Divergence Distance by counting the pixel value distribution channel by channel. Additionally, we also compare the differences between the three standard color charts in each image pair and use MAE for measurement.

**Testing Results.** The quantitative results are presented in **Table** 2. Chai et al.'s uniform transformation-based method has shown poor performance in terms of the KL metric, thus affirming the advantages of our patch-wise method for scenes with multiple illumination conditions. **Fig.** 5 shows the visualization results of Chai et al.'s and ours.

### 4.3 Cross Camera ISP

**Experiment Design.** To evaluate the ability of the raw-to-raw model to generalize the ISP model across different cameras, we design the following experiment which treat raw-to-raw model as

**Table 2: Quantitative results of multi illumination raw-to-raw mapping task. Bold indicates the best result.**

| Model | Sony→Galaxy | | Galaxy→Sony | |
|---|---|---|---|---|
| | MAE↓ | KL↓ | MAE↓ | KL↓ |
| Cyclegan [41] | 0.055 | 1.06 | 0.029 | 0.78 |
| Cut [29] | 0.040 | 0.86 | 0.028 | 0.79 |
| Swin-UNIT [23] | 0.045 | 1.37 | 0.046 | 1.06 |
| Chai et al.'s [9] | 0.038 | 1.59 | 0.031 | 1.62 |
| UVCGAN [36] | 0.048 | 1.03 | 0.037 | 0.87 |
| Ours | **0.037** | **0.74** | **0.028** | **0.59** |

domain adaptation module or data augmentation step as illustrated in **Fig.** 1: (b) (c).

In this experiment, we apply raw-to-raw models trained in **Sec.** 4.1, including UVCGAN, Afifi et al.'s, and our model, before various Deep ISP models[15–18] in their training or testing phase. We denote the raw images captured by iPhone-x and their rendered sRGB images as image pair $A \rightarrow A$, and similarly, those captured and rendered by Samsung-s9 as $B \rightarrow B$.

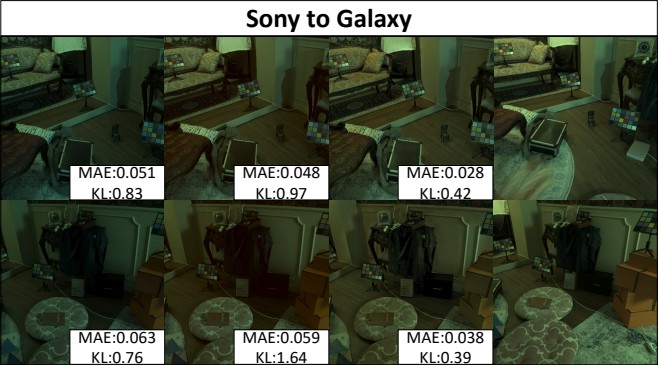 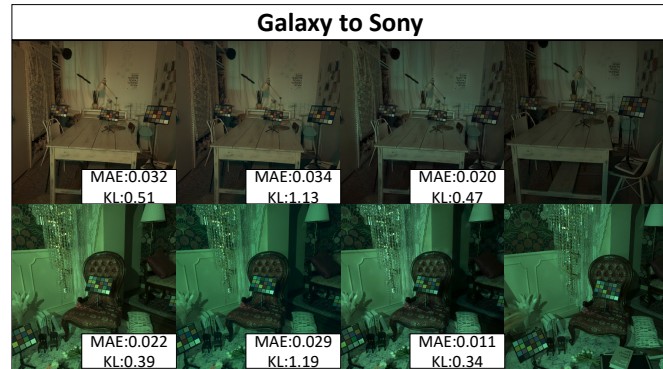

Figure 5: Qualitative results of multi-illumination raw-to-raw task. Each block, from left to right, represents the source camera, Chai et al.'s [9], our method, and ground truth, respectively. The 1/1.6 gamma correction is also applied for visualization.

Specifically, for the evaluation of the data augmentation step, we use the raw-to-raw model to convert the raw images captured by iPhone-x to the color space of Samsung-s9, while keeping the sRGB images unchanged, resulting in a new image pair $AB \rightarrow A$. We train the Deep ISP model using this image pair and test it on $B \rightarrow A$. It's worth noting that we don't test on $B \rightarrow B$ because sRGB images rendered by different ISP also exhibit significant differences. For the evaluation of the adaptation module, we first train the Deep ISP model using image pair $A \rightarrow A$ and then testing by $BA \rightarrow A$. We then swap two cameras and repeat the experiments. In this experiment, there is no overlap between the training set of the raw-to-raw model and the test set of the ISP task.

Due to the fact that $A \rightarrow B$ and $B \rightarrow A$ are scene-level paired rather than pixel-level paired, the KL Divergence Distance (see **Eq.** 14) is also used as a metric. Additionally, we employ the FID [30] (Fréchet Inception Distance) metric which commonly used in generative adversarial networks evaluation, to measure the consistency between the generated image set and the ground truth set in high-dimensional space.

**Testing Results.** Cross Camera ISP is a highly challenging task, even minor differences in the raw domain can become noticeable due to the non-linear operations of the ISP. The quantitative results presented in **Table** 5 demonstrate that our proposed model can effectively enhance the performance of Deep ISP, whether when it is deployed in training or testing phase. Compared with the Afifi et al.'s semi-supervised method, our unsupervised method performs better in FID metric and comparable in KL metric. As shown in in **Fig.** 6, our visualization result surpass the other methods significantly. Overall, our results exhibit closer proximity to the ground truth and exhibit fewer instances of local color loss, this further underscores the effectiveness of our parameterired model and patch-wise transform strategy.

## 5 ABLATION STUDY

We demonstrate the effectiveness of the CSTPP module in raw-to-raw task through following ablation experiment. Since the backbone of our proposed model is not a generative model, we use the UVCGAN [36] for this experiment. Specifically, we replace the decoder part of UVCGAN with the CSTPP module. Subsequently, we conduct the same experiment as mentioned in **Sec.** 4.1, and the quantitative results are shown in **Table** 3. The inclusion of the

CSTPP module significantly improves the performance of uvcgan, especially a notable increase in the SSIM metric. This indicates that the channel-level combination performed by the CSTPP module can effectively preserve the structural information of the image, without causing local deformation or introducing artifacts.

Table 3: Result of ablation study for the CSTPP module.

| Method | Samsung-s9→iPhone-x | | | iPhone-x→Samsung-s9 | | |
|---|---|---|---|---|---|---|
| | PSNR↑ | SSIM↑ | MAE↓ | PSNR↑ | SSIM↑ | MAE↓ |
| W/o CSTPP | 27.22 | 0.82 | 0.031 | 26.10 | 0.79 | 0.037 |
| W/ CSTPP | 28.84 | 0.89 | 0.027 | 26.49 | 0.85 | 0.037 |

Additionally, since our proposed model is parameterized, we can significantly reduce computational complexity by downsampling input images multiple times. We conducted downsampling from 2 to 16 times during the testing phase, and the quantified results are shown in **Table** 4. Even after reducing the pixel count of input image by hurdreds times (16× downsample), our model still maintains good performance.

Table 4: Result of ablation study for downsample times.

| Downsample Times | Samsung-s9→iPhone-x | | | iPhone-x→Samsung-s9 | | |
|---|---|---|---|---|---|---|
| | PSNR↑ | SSIM↑ | MAE↓ | PSNR↑ | SSIM↑ | MAE↓ |
| 16× | 29.00 | 0.88 | 0.028 | 27.97 | 0.89 | 0.033 |
| 8× | 29.04 | 0.89 | 0.028 | 27.95 | 0.88 | 0.034 |
| 4× | 29.02 | 0.89 | 0.027 | 27.79 | 0.88 | 0.034 |
| 2× | 29.20 | 0.89 | 0.027 | 27.89 | 0.88 | 0.034 |
| 1× | 29.73 | 0.90 | 0.025 | 28.09 | 0.89 | 0.033 |

## 6 CONCLUSION

In this paper, we propose an unsupervised raw-to-raw model for the first time and introduce it to address the generalization issue of ISP. Our parameterized model directly predicts color space transformation parameters in a patch-wise manner, enabling accurate and flexible handling of the raw-to-raw task in general scenes. Extensive experimental results demonstrate that our model excels not only in the raw-to-raw task but also in achieving cross-camera deployment for ISP. Considering the simplicity and effectiveness of our proposed method, it has great potential to be directly deployed into exisintg AI-ISP platforms, which is one of our ongoing work.

 

**Table 5: Quantitative results of the cross-camera ISP task. Results were obtained by training and testing four Deep ISP models: PyNET[18], microISP[17], PyNET-v2[16], and syenet[15]. The mean performances of the four ISP models are presented, detailed results are provided in the supplementary material. Bold indicates the best result.**

| ISP Model | R2r Model | Deploy iPhone-x ISP After Samsung-s9 Sensor | | | | Deploy Samsung-s9 ISP After iPhone-x Sensor | | | |
|---|---|---|---|---|---|---|---|---|---|
| | | Train Set | Test Set | KL↓ | FID↓ | Train Set | Test Set | KL↓ | FID↓ |
| Average | W/o R2r | $A \rightarrow A$ | $B \rightarrow A$ | 1.43 | 80.93 | $B \rightarrow B$ | $A \rightarrow B$ | 2.19 | 83.04 |
| | UVCGAN [36] | $AB \rightarrow A$ | $B \rightarrow A$ | 1.18 | 93.68 | $BA \rightarrow B$ | $A \rightarrow B$ | 0.98 | 86.66 |
| | | $A \rightarrow A$ | $BA \rightarrow A$ | 1.00 | 116.96 | $B \rightarrow B$ | $AB \rightarrow B$ | 0.89 | 99.11 |
| | Afifi et al.'s [2] | $AB \rightarrow A$ | $B \rightarrow A$ | 1.06 | 67.01 | $BA \rightarrow B$ | $A \rightarrow B$ | **0.73** | 77.94 |
| | | $A \rightarrow A$ | $BA \rightarrow A$ | 1.08 | 74.68 | $B \rightarrow B$ | $AB \rightarrow B$ | **0.77** | 79.14 |
| | Ours | $AB \rightarrow A$ | $B \rightarrow A$ | **1.03** | **61.29** | $BA \rightarrow B$ | $A \rightarrow B$ | 0.80 | **69.51** |
| | | $A \rightarrow A$ | $BA \rightarrow A$ | **0.97** | **62.53** | $B \rightarrow B$ | $AB \rightarrow B$ | 0.82 | **74.00** |

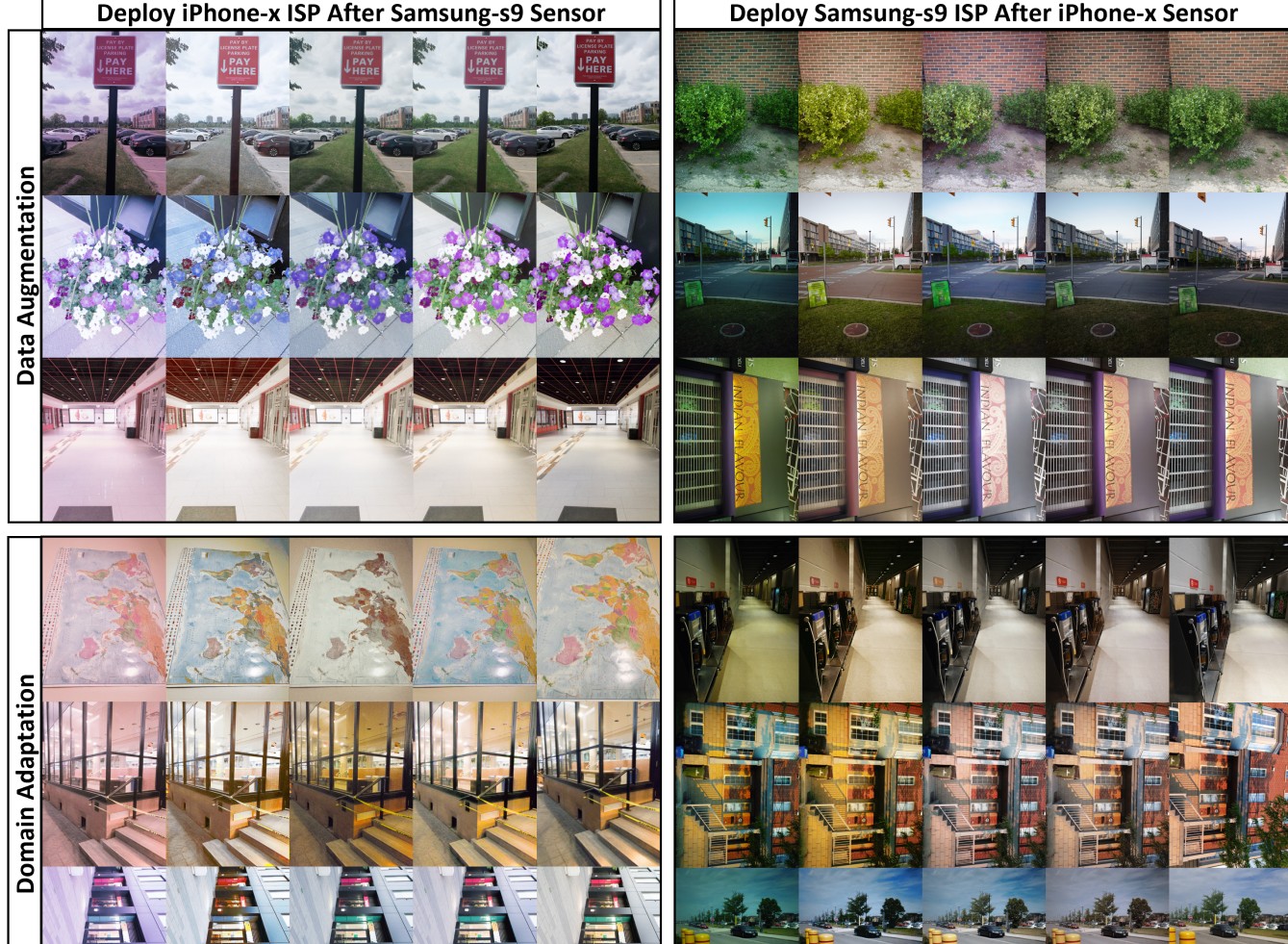

**Figure 6: Qualitative results of cross-camera ISP task. Each block, from left to right, represents the W/o r2r, UVCGAN [36], Afifi et al.'s [2], our method, and ground truth, respectively.**

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
