# OpenReview forum: "Generalizing ISP Model by Unsupervised Raw-to-raw Mapping"
_acmmm.org/ACMMM/2024/Conference — MM2024 Oral_

### Official Review · Reviewer_7QRE · 2024-05-23

**Rating:** 5
**Confidence:** 3

**Summary:**

This paper proposes a CycleGAN-based framework for unsupervised RAW-to-RAW mappings, where a color space transformation parameters predictor module is proposed for accurate color mapping. Experiments demonstrate the effectiveness of the proposed method.

**Strengths:**

(1). The overall framework is well-designed and the proposed ideas are well-motivated.

(2). The overall writing is good.

(3). Experiments and analyses are thorough throughout the paper.

**Limitations:**

(1). I notice that the input images are not fully aligned with the corresponding reference images in Fig 4 and Fig. 5, so do the authors calculate PSNR, SSIM, and MAE directly using the reference image or after aligning the reference image with the input image?

(2). The authors should compare the computational efficiency of the proposed method with other methods in terms of inference speed, model size, and memory consumption.

**Suitability:**

3

---

### Official Review · Reviewer_dxjJ · 2024-05-24

**Rating:** 5
**Confidence:** 3

**Summary:**

The paper proposes an approach to address the challenge of Image Signal Processor (ISP) generalization across different cameras by introducing an unsupervised raw-to-raw mapping model. The authors develop a Color Space Transformation Parameters Predictor (CSTPP) module that predicts color space transformation parameters in a patch-wise manner and integrates this with a CycleGAN-style training framework. The results demonstrate the effectiveness of the proposed method.

**Strengths:**

1. The design is reasonable, predicting a patch-wise mapping matrix between two domains.
2. The performance appears promising, showing better results than a semi-supervised method published in BMVC 2021.

**Limitations:**

1. The main issue of the paper is the writing quality in my opinion.
2. The writing of the paper needs improvement. The introduction of the significance of this task is not very clear, especially for those unfamiliar with it. For instance, one might wonder why not use the color matrix saved in metadata to obtain a relatively camera-independent color space first. Providing a discussion supported by experiments would be better.-
3. Many figures are unclear. For example, Fig. 2 seems meaningless and lacks sufficient explanation. It appears that the mainstream of existing methods also involves learning the mapping matrix instead of direct image-to-image mapping.
4. Eq. 7 is somewhat confusing. What is the meaning of “implying that Eq. 6 only involves a single row” in L327?

**Suitability:**

2

---

### Official Review · Reviewer_GoXm · 2024-05-26

**Rating:** 4
**Confidence:** 4

**Summary:**

The paper is well-written and presents a significant contribution to the field of image processing. The proposed method shows promise, particularly in its ability to handle the raw-to-raw mapping task in an unsupervised manner. However, the paper could be improved by addressing the potential weaknesses, such as providing a more thorough comparison with existing methods, discussing the model's limitations, and demonstrating its practical utility in real-world applications. Despite these areas for improvement, the paper is a valuable addition to the literature and offers a solid foundation for future research in this area.

**Strengths:**

1.The introduction of an unsupervised raw-to-raw model and the CSTPP module is a significant advancement in the ISP field.
2.The model demonstrates comparable performance to state-of-the-art semi-supervised methods in both raw-to-raw and cross-camera ISP tasks.
3.The method has potential for direct deployment in existing AI-ISP platforms, highlighting its real-world applicability.

**Limitations:**

1. More detailed descriptions of the CSTPP module and the CycleGAN-style training framework are needed to facilitate replication and deeper comprehension. Specifically, it would be beneficial to provide the motivations behind these components.
2. Please include the corresponding comparison mentioned in related work in the experimental results . Additionally, the paper could benefit from a more detailed comparison with state-of-the-art methods, particularly regarding computational efficiency and practical applicability.
3. The paper cites the Conformer model developed by others; please describe it in more detail so that the reader can easily understand its function and relevance.
4. The ablation experiments need to be refined. For instance, Please conduct an ablation experiment on the Conformer. It is important to demonstrate that the validity of the method stems from your proposed module and not solely from the Conformer.
5. Pay attention to the formatting and writting..

**Suitability:**

3

---

### Meta-Review · Area_Chair_PnMZ · 2024-07-06

**Recommendation:** Accept (Oral)
**Confidence:** 4

**Metareview:**

The paper received positive ratings (BA, WA, WA) from the reviewers pre-rebuttal. The reviewers raised a few concerns and suggestions, which were duly addresses in the rebuttal. Two reviewers further raised their ratings post rebuttal, thereby resulting in final ratings of WA, WA and A respectively. The AC agrees with the reviewers' positive evaluation of this work and recommends acceptance.